# Assessing mPTC Progression during Active Surveillance: Volume or Diameter Increase?

**DOI:** 10.3390/jcm10184068

**Published:** 2021-09-09

**Authors:** Maria Cristina Campopiano, Antonio Matrone, Teresa Rago, Maria Scutari, Alessandro Prete, Laura Agate, Paolo Piaggi, Rossella Elisei, Eleonora Molinaro

**Affiliations:** Unit of Endocrinology, Department of Clinical and Experimental Medicine, University of Pisa, 56124 Pisa, Italy; cristina.campopiano@gmail.com (M.C.C.); anto.matrone@yahoo.com (A.M.); rago@endoc.med.unipi.it (T.R.); mariascutari@hotmail.it (M.S.); alessandro.prete22@gmail.com (A.P.); laura.agate@virgilio.it (L.A.); paolo.piaggi@gmail.com (P.P.); elemoli@hotmail.com (E.M.)

**Keywords:** differentiated thyroid cancer, papillary thyroid microcarcinoma, active surveillance, management, observation, tailored treatment for thyroid cancer

## Abstract

Active surveillance (AS) is considered an alternative to immediate surgery in micropapillary thyroid carcinoma (mPTC). However, the definition of clinical mPTC progression during AS is controversial. We evaluated changes in tumor size using both tumor diameters and volume in 109 patients with mPTC followed in an AS protocol for a mean period of 31 ± 18 months. At the time of data lock, 19/109 (17.4%) mPTC reached and maintained a volume increase of ≥50%. However, only 3/19 (15.7%) showed progression, according to the diameter increase. The remaining 16 showed a slight diameter growth without reaching the original protocol progression criteria. The mean mPTC growth rate in stable cases was 0.37 mm^3^/month, while it was significantly greater in the mPTC, which achieved a volume change ≥50% with respect to the other. The two mPTC that developed a significant diameter increase had a growth rate of 41 and 18 mm^3^/month. Instead, the growth rates of the three mPTC that developed lymph node metastases were 0, 2.5 and 16 mm^3^/month. The ≥50% volume increase appears to be a too sensitive marker of disease progression, with a downstream higher surgery rate. The assessment of growth rate could distinguish mPTC with high and low growth rates, which would allow us to tailor the algorithm of the evaluations to a more appropriate timing.

## 1. Introduction

Active surveillance (AS) is a watchful waiting approach that allows one to closely monitor a patient’s condition without treatment until clinical disease is overt [1]. It is the only available way to plan therapies and their timing, avoiding either under- or over-treatments. Recently, AS has been proposed as an alternative to surgery in unifocal and intrathyroidal papillary microcarcinoma (mPTC) [2,3,4] and it is primarily indicated to old or frail mPTC patients [2,3]. The indolent nature of disease, demonstrated in several clinical trials [5,6], poses no major threats to patients and the delay of surgery appears to be safe even when local metastases develop. Since then, observational studies from different countries in Asia, America and Europe have confirmed the favorable outcome of mPTC during AS [7,8,9,10,11,12,13,14,15,16], underlining that it is related to the indolent course of the disease and not to the efficacy of active treatment.

However, there is no clear consensus on the definition of clinical mPTC progression during AS. The criteria of mPTC growth >3 mm in each diameter and/or the appearance of metastatic lymph node may not practically reflect the real aggressiveness of the tumor. While the appearance of lymph node metastases could be an unequivocal sign of progression, tumor growth is arbitrarily defined based on experience in the field [5,6,15]. Some authors speculated that the increase in mPTC volume allowed them to earlier identify those tumors that would grow at higher speed and on its turn may be more aggressive and clinically significant [9,11]. Others, however, have argued that the evaluation of tumor volume could be too sensitive and still shift to surgery in many cases [17,18].

Advances in knowledge on tumor growth could help in understanding which patients could benefit from more careful monitoring and better define the ideal frequency of evaluations and the optimal timing for surgery. In our previous report [15], we set the cut-off of disease progression at a growth of 3 mm or more in each diameter in two consecutive echo assessments, six months apart. The aim of this study was to analyze the mPTC growth by comparing the increase of 3 mm in each diameter with the volume increase to define a personalized surveillance protocol based on tumor growth.

## 2. Patients and Methods

### 2.1. Patients

We identified a cohort of 127 patients with cytological diagnosis or suspicious papillary thyroid cancer (PTC) measuring ≤1.3 cm at neck ultrasound (nUS), prospectively enrolled and followed in AS protocol at the University Hospital of Pisa, Italy, from November 2014 to November 2020. A total of 18/127 were excluded from this analysis because the period of observation was <6 months. The inclusion criteria in the original protocol were described in our previous report [15].

All patients who agreed to participate in this program signed an informed consent at study entry. The appearance of metastatic lymph nodes, confirmed by a cytology and thyroglobulin (Tg) measurement on the wash-out fluid of the needle used for FNAC, or an increase in size more than 3 mm for each nUS diameter of mPTC, confirmed in two consecutive examinations, defined the mPTC progression according to the original protocol criteria and were considered indications to transition from AS to surgical intervention. Patients were regularly evaluated every 6 months for the first 2 years and then yearly. Levothyroxine (LT4) therapy was administered, or maintained, in hypothyroid patients to obtain a thyroid-stimulating hormone (TSH) level within the normal range. This study (number 334/2014) was approved on 20 November 2014 by the Local Ethical Committee (Comitato Etico Area Vasta Nord-Ovest-CEAVNO). 

### 2.2. Methods

#### Neck Ultrasound

nUS was performed using a real time instrument (Esaote SPA, Genova, Italy; My Lab 50 machine with 7.5–12 MHz linear transducer). During the follow-up, nodules and suspicious lymph nodes in neck stations were inspected. nUS was performed by the same independent US-trained endocrinologists (E.M. and M.C.C.). Accurate descriptions of echogenicity, microcalcifications, integrity of halo, lengths of antero-posterior (AP), latero-lateral (LL) and longitudinal (Long) diameters were recorded in a computerized database. 

To the purpose of the present study, we calculated the nodular volume using the ellipsoid formula (AP*LL*Long/0.52); therefore, we looked at the change in volume at each control (V2) with respect to the baseline volume (V1), expressed as a percentage and using the mathematical formula (V2-V1)/V1*100. According to previous studies, including [9,11], a meaningful change in tumor volume was defined as when there is a size increase of ≥50% compared to baseline values. We also calculated the growth rate of mPTC, expressed in mm^3^/month, as the slope of the regression line between the volumes calculated at each visit and the time of AS.

### 2.3. Statistical Analysis

The normality of the variables was tested by means of the Shapiro–Wilk test. Data are expressed as mean ± standard deviation (variables with normal distribution) or as median with interquartile range (variables with non-Gaussian distribution). The differences between groups for continuous variables with normal distribution were evaluated using the t-Student test (two groups) after evaluating the homogeneity of the variances of the groups using the Levene test. Differences between groups for continuous variables with non-Gaussian distribution were compared by means of the Mann–Whitney U test (two groups). Associations between categorical variables were assessed with the chi-square test or Fisher’s test where appropriate. All statistical analyses were conducted with SPSS software (IBM SPSS Statistics, Armonk, New York, NY, USA; version 25).

## 3. Results

In this analysis, we included 109 mPTC patients who were prospectively observed for a mean of 31 ± 18 months.

During AS, only 5/109 patients (4.5%) showed mPTC progression, according to the original protocol criteria [15]. In two patients, all three mPTC diameters increased by 3 mm in two 6-month consecutive US evaluations, while in three mPTC patients, lymph node metastases developed during AS. The epidemiological, ultrasonographic and pathological characteristics of progressing cases are summarized in Table 1. Of these latter, only one showed the diameter increase but not enough to fulfil the criteria of the original protocol. These patients developed mPTC progression after a median period of observation of 18 months (IQR 14–25), which was significantly shorter than the median time of observation of the other stable mPTC (32 months, IQR 21–48) (*p* = 0.04).

Regarding the increase of the volume of mPTC, we found that during the AS, 22/109 (20%) patients showed an increase in mPTC volume of ≥50% compared to baseline at any time during the observation. As shown in Table 2, most of the patients showed a volume increase ≥50% at 6-, 12- and 18-month visits (18/22); only 4/22 experienced a volume increase >50% after a 24-month visit. In 19/22 (17%) cases, the volume increase of >50% was confirmed at the last evaluation; meanwhile, the remaining three showed a volume reduction within the 6 following months.

As shown in Figure 1, based on the change in volume at the last evaluation, 90/109 (83%) cases had a volume variation <50% (Group A) and 19/109 (17%) (Group B) had a volume variation ≥50% (range 50–400%), compared to baseline. According to previous studies (10–12), Group B represents the “growing group” and 3/5 progressing cases for the original protocol (i.e., two cases for the increase of diameters and one case for the development of lymph node metastases) belonged to this latter group. In terms of variance, the other two cases that developed lymph node metastases did not show a meaningful volume increase. 

As shown in Table 3, in our series, there were no differences in the epidemiological, clinical, biochemical, US and cytological characteristics between patients of Group A and Group B.

The follow-up of Group B and Group A was similar (32 months vs. 30 months, respectively). However, the timing during which the volume change occurred varied from 6 to 48 months. Because of this difference we calculated the growth rate that was 1.1 mm^3^/month in the entire series (*n* = 109), 6.56 mm^3^/month in Group B and 0 mm^3^/month in Group A. As shown in Figure 2, the comparison of the volume growth rates of the two groups showed a statistically significant difference (*p* < 0.0000001).

The volume growth rate of all our mPTC with the progression calculated according to the criteria of the original protocol was then compared. As shown in Figure 3, the mPTC growth rate was near to 0 (mean 0.37 mm^3^/month, ±SD 7.9 mm^3^/month) in stable cases. The two mPTC that progressed according to the original protocol criteria had a growth rate of 41 and 18 mm^3^/month. Instead, the growth rates of the three mPTC that developed lymph node metastases were 0, 2.5 and 16 mm^3^/month. This latter was the one in whom there was a simultaneous increase of diameters but not sufficient to fulfil the pre-defined criteria [15].

## 4. Discussion

Active surveillance has been proposed as an alternative strategy to immediate surgery in low-risk mPTC [2,3,4] and it is recommended for old or frail mPTC patients [2,3]. The mPTC progression during AS is the main reason for interrupting AS and shifting to surgery. There is no univocal definition of mPTC progression during AS. While the appearance of lymph nodes or distant metastases is a clear sign of progression [6,8,9,10,14,15], the evaluation of tumor progression is quite different in different centers [6,8,9,10,11,14,15]. The major controversial issue is if it is more appropriate to consider the linear increase of diameters or the volume increase.

In the present study we found that 19/109 (17.4%) of our mPTC patients had a persistent volume increase ≥50% at the last evaluation, which would have been considered meaningful by several authors [9,11]. According to these criteria, these 19 patients would have been submitted to surgery in a timeframe varying from 6 to 48 months, while 16 of them remained in AS because they did not show the progression according to the pre-defined criteria [15]. Moreover, 18 cases showed the volume increase ≥50% within 24 months, while only 5 cases progressed in this timeframe when considering the pre-defined criteria [15]. According to these findings, it appears that many more mPTC cases would be sent to surgery when volume increase is considered, thus reducing the real impact of the AS. In this context, more attention should be paid to smaller mPTC (i.e., 4–5 mm), because any even insignificant variation in diameters (1–2 mm) might cause a volume increase >50%, which could be misread as progression. Instead, it could likely reflect the variability in US measurements [19]. Besides, the increase of 3 mm or more in very small mPTC may not be clinically relevant and these patients may continue AS [6].

Nevertheless, and in agreement with the results of Tuttle et al. [9], both mPTC cases that enter progression because of the 3 mm diameters increase, also showed a volume increase ≥50%. This was not the case of the three mPTC progressing for the development of lymph node metastases since two of them did not reach the volume increase ≥50%, thus the appearance of cervical lymph node metastases must be a criterion for surgery independent from the tumor growth, either considered as diameter or volume increase. Following this criterion, the delayed surgery needs to be more extensive than the immediate one; however, it is still safe in term of disease outcome and surgery adverse events. Moreover, this strategy allows us to operate only the mPTC patients who need treatment, avoiding unnecessary surgery procedures.

No differences were found among the epidemiological, clinical, US and cytological characteristics between the two groups of patients divided according to the volume variation of ≥50% or <50%. Unfortunately, we were unable to find any prognostic factors of progression, neither in this study considering the increase of volume nor in the previous one using diameter as a parameter of growth [15]. Probably in the future, a characteristic molecular pattern could discriminate the risk of mPTC progression during AS. Currently, the role of BRAF mutations and TERT promoter rearrangements is not fully understood in mPTC during AS, although they strongly predict a poor prognosis in operated DTC [20]. Other gene mutations could likely give to cancer cells a selective advantage for progression. Recently, mutations of genes coding for cell adhesion molecules or migration proteins seem to discriminate mPTC with greater aggressiveness [21].

It is known that the tumor growth is non-linear by definition and that periods of growth can be followed by periods of shrinkage or stabilization [17]. The evidence of nodular growth in two consecutive US evaluations should minimize the risk of an inconstant growth and send to surgery a nodule that is really progressing. Assessing tumor growth in two different occasions, we can also reduce intra- and inter-observer variability at US [19].

According to our results, it appears that it is very relevant to consider how long this variation takes: it is plausible that reaching a volume increase ≥50% compared to the baseline in 6 months or 4 years had a different clinical relevance and the evaluation of the growth rate of mPTC could solve this aspect. In our series, almost all mPTC had a growth rate near to 0 mm^3^/month and this data is consistent with the low progression rate of mPTC during AS. However, progressing mPTC patients who developed diameters increase ≥3 mm had a faster growth rate than stable ones and those who showed lymph node metastases had a variable growth rate, ranging between 0 and 16 mm^3^/month. Our analysis highlights a different behavior in growth rate between mPTC that developed lymph node metastases from those that had an increase in diameters. On the one hand, the assessment of the growth rate could identify a subgroup of patients with mPTC at risk of progression, while on the other hand it cannot exclude the risk of developing lymph node metastases. This evidence makes essential the use of US and it is fundamental that during AS the cervical lymph nodes and not only the thyroid gland, are regularly evaluated.

Notably, in our original cohort, the mPTC progression developed in all five cases within two years after the enrolment, while other mPTC remained stable after a mean follow-up of almost 3 years. This result could represent a surrogate marker of the growth rate of mPTC and we can hypothesize that an mPTC that enters progression in a short time is likely a rapidly progressing mPTC destined to become a clinically relevant disease.

Although larger data analyzing the evolution of mPTC are needed to better define the most appropriate timing for surgery, the assessment of the growth rate of the mPTC could help us identify the minority of mPTC with a higher growth rate and thus having a greater risk of progression during AS. This subgroup could benefit from a more stringent observation with more frequent evaluations or, conversely, the other group could be less frequently surveyed. We might plan a different algorithm, according to mPTC growth rate, with more frequent evaluations in those with a higher growth rate than in those with a lower growth rate. Further studies will be helpful to determine the clinical significance of growth rate in mPTC diameter and volume and refine the thresholds for intervention. To date, the indication to surgery remains the same of previous reports based on diameter increase [15]. The objective of future research will be to find biomarkers able to early identify mPTC that tend to progress and perform less extensive therapeutic approach. One of the limitations to the present study is that a real 3D imaging technique was not used. It is possible that a more refined technique such as MRI, even if much more expensive, may improve the ability to objectively assess clinically meaningful growth, albeit minimal.

In conclusion, our study showed that, considering volume variation, to define mPTC progression, without taking into account the initial size and the period in which volume increase occurs, it can induced to operate in many more cases that could have benefited from a longer AS. Indeed, the 3 mm diameter increase in two consecutive evaluations seems, at the moment, the best trade-off in assessing tumor progression that should be matched by clinical progression.

## Figures and Tables

**Figure 1 jcm-10-04068-f001:**
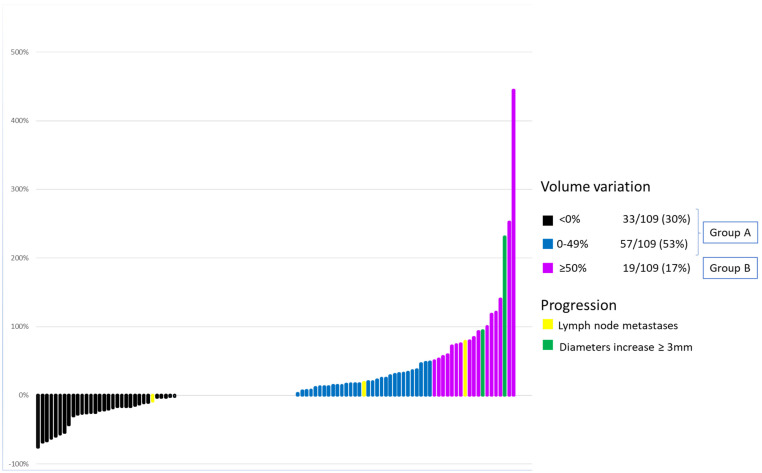
Each line expresses the extent of the volumetric variation of mPTC at the last check with respect to enrollment. Yellow and green lines represent the mPTC that have developed clinical progression during observation according to protocol criteria. Green lines identify the 2 mPTC that showed a ≥3 mm increase in each diameter. Yellow lines identify the 5 mPTC that have developed lymph node metastases.

**Figure 2 jcm-10-04068-f002:**
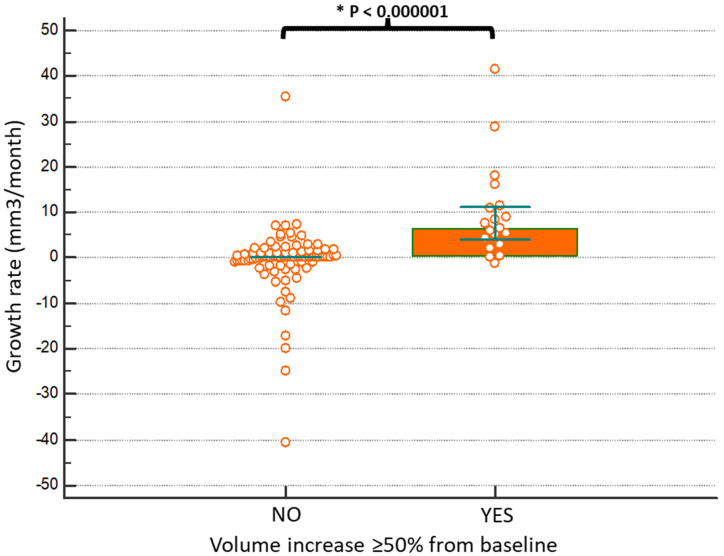
Patients who had a volumetric change in mPTC greater than 50% compared to baseline at the last check-up showed a greater speed of growth (6.56 versus 0 mm^3^/month).

**Figure 3 jcm-10-04068-f003:**
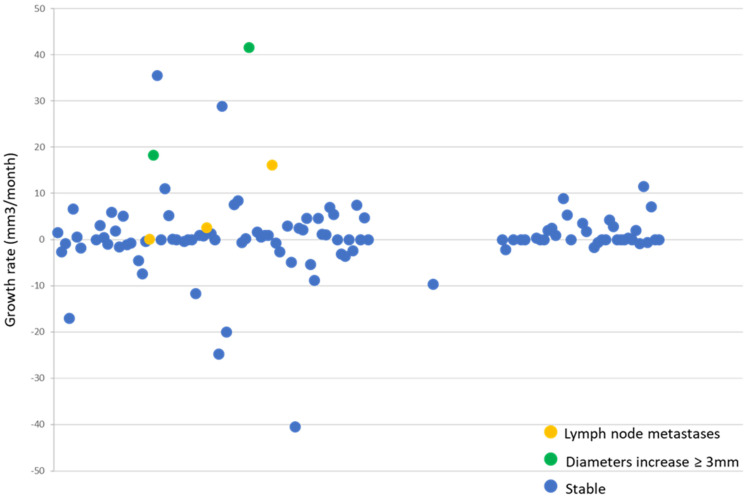
Each dot corresponds to a patient. The growth rate is expressed on the ordinates axis. The green dots represent the progressing mPTC patients by each diameters increase >3 mm, the yellow ones represent the progressing mPTC who developed lymph node metastases and the blue ones represent the stable mPTC patients. The mean growth rate is near 0 mm^3^/month. The green dots have the fastest growth rate, the yellow ones have variable behavior in tumor growth and the blue ones have the slowest growth rate.

**Table 1 jcm-10-04068-t001:** Epidemiological, clinical and pathological features of progressing mPTC.

	Patient A	Patient B	Patient C	Patient D	Patient E
Age at diagnosis	51 years	36 years	45 years	36 years	38 years
Gender	Female	Female	M	Female	Female
Kind of progression	LNF MTS	mPTC growth	mPTC growth	LNF MTS	LNF MTS
mPTC maximum diameter at baseline	9 mm	13 mm	13 mm	12 mm	13 mm
mPTC maximum diameter at last evaluation	9 mm	18 mm	18 mm	11 mm	13 mm
mPTC volume at baseline	220 mm^3^	1100 mm^3^	800 mm^3^	560 mm^3^	670 mm^3^
mPTC volume at last evaluation	260 mm^3^	2100 mm^3^	1600 mm^3^	520 mm^3^	880 mm^3^
Period of observation	12 months	13 months	23 months	36 months	24 months
Kind of surgery	TT + LC LNX	TT	TT	TT + LC LNX	TT + LC LNX
TMN (8° edition)	T1b(m)N1bMX	T1bNXMX	T1bNXMX	T1b(m)N1bMX	T1b(m)N1bMX
Histothype	Tall cell	Classical	Classical	Classical	Classical
mPTC maximum diameter at histology	13 mm	17 mm	17 mm	15 mm	13 mm
Minimal extrathyroidal extension	No	No	No	Yes	No
Maximum diameter of LNF MTS	5 mm	-	-	5 mm	3 mm
Radioiodine therapy (activity)	Yes (30 mCi)	No	No	Yes (100 mCi)	Yes (130 mCi)
Follow-up after surgery	36 months	36 months	18 months	12 months	18 months
Outcome	Excellent response	Excellent response	Excellent response	Excellent response	Excellent response

LNF MTS: lymph node metastases, TT: total Thyroidectomy, LC LNX: latero-cervical lymphadenectomy.

**Table 2 jcm-10-04068-t002:** Volume increase ≥ 50% at different time evaluations.

Period of Observation (Months)	Volume Increase > 50% (*n*)	Total (*n*)	%
+6	8	109	7.3
+12	6	87	6.9
+18	4	66	6.1
+24	1	54	1.9
+36	2	42	4.8
+48	1	30	3.3

**Table 3 jcm-10-04068-t003:** Comparison of clinical, ultrasound and cytological features in patients showing less or more than 50% of the volume variation from baseline to last evaluation.

Clinical, Ultrasound and Cytological Features	Volume Variation < 50% (*n* 90) Group A	Volume Variation ≥ 50% (*n* 19) Group B	*p*
Gender Female *n*(%) Male *n*(%)	66 (73%) 24 (27%)	13 (68%) 6 (32%)	NS
Age (years) Media ±DS	44 ± 15	45 ± 10	NS
Age < 40 years	33 (37%)	7 (37%)	NS
Hereditary for PTC *n*(%)	5 (6%)	3 (15%)	NS
Previous external beam radiation *n*(%)	3 (3.3%)	0	NS
Hypothyroidism treated with levothyroxin *n*(%)	11 (12%)	2 (11%)	NS
TSH baseline (mUI/l) mean ± SD	1.6 ± 0.8	1.5 ± 0.8	NS
Tg baseline (mcg/l) mean ± SD ^	10.9 ± 10.1	10.9 ± 11.4	NS
AbTg baseline (UI/mL) mean ± SD	2.2 ± 4.7	1.6 ± 4.7	NS
Cytological results TIR4 *n*(%) TIR5 *n*(%)	40 (44%) 50 (56%)	7 (37%) 12 (63%)	NS
Hypoechogenicity *n*(%)	80 (89%)	19 (100%)	NS
Microcalcification *n*(%)	48 (53%)	10 (53%)	NS
Irregular margins *n*(%)	58 (64%)	11 (58%)	NS
Thyroid capsular proximity *n*(%) *	47 (52%)	12 (63%)	NS
Maximum diameter baseline mm mean ± SD	9.1 ± 24	9 ± 2.5	NS
Nodular volume at baseline mm^3^ mean ± SD	305 ± 214	262 ± 216	NS

PTC: papillary thyroid cancer, TSH: thyrotropin, Tg: thyroglobulin; * Thyroid capsular proximity is defined as a lesion located near the thyroid capsule, within 3 mm abutting the capsule; ^ We evaluated Tg values in patients without Tg antibodies interferences (87/109).

## Data Availability

The data presented in this study are available on request from the corresponding author. The data are not publicly available due to patient privacy and the General Data Protection Regulation.

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
