# Peer review of "Assessing mPTC Progression during Active Surveillance: Volume or Diameter Increase?"

_jcm, 2021, doi:10.3390/jcm10184068_

Round 1
Reviewer 1 Report
Major Comments
Dear authors,
Your article is interesting.
Nevertheless, some aspects require refinement:
- Measurement of anti-Tg antibodies should be included.
- In Figure 3, you mention that: „Almost all have a growth rate of 0. The green dots have the fastest growth rate, the yellow ones have variable behavior in tumor growth.” Please explain or rephrase in order to clarify which group of mPTC patients essentially lacks a growth rate.
- You mention that: “It is possible that a more refined technique such as MRI may improve the ability to assess a significant growth.” Yet, please note that as compared with nUS, MRI, albeit very expensive, is more objective and would thus be useful in order to assess a minute but clinically meaningful rather than a significant growth.
Minor Comments/Clarifications with suggested alterations
Some points should be improved in terms of English language such as the following:
- Please note that “familiarity” probably does not express the intended meaning (Table 1 and Table 4). Please explain or rephrase.
- Please note that the sentence:
“The mean mPTC growth rate in stable cases was 0.37 mm3/month while it was significantly greater in the mPTC which achieves a volume change ≥50% respect to the other.“ might be better rephrased as:
“The mean mPTC growth rate in stable cases was 0.37 mm3/month while it was significantly greater in the mPTC which achieves a volume change ≥50% with respect to the other.“
Author Response
We thank the reviewer for their thoughtful criticism. In bold their comments and in italic our responses.
Dear authors,
Your article is interesting.
Nevertheless, some aspects require refinement:
Major Comments
- Measurement of anti-Tg antibodies should be included.
Thank you for the advice. We added the measurement of anti-Tg antibodies in Table 3 of the revised version of the manuscript. Please note that we evaluated Tg values after the exclusion of AbTg interferences, as stated in the Table 3 footnote
- In Figure 3, you mention that: „Almost all have a growth rate of 0. The green dots have the fastest growth rate, the yellow ones have variable behavior in tumor growth.” Please explain or rephrase in order to clarify which group of mPTC patients essentially lacks a growth rate.
Thank you. We have rephrased according to your suggestion (Lines 161-162)
- You mention that: “It is possible that a more refined technique such as MRI may improve the ability to assess a significant growth.” Yet, please note that as compared with nUS, MRI, albeit very expensive, is more objective and would thus be useful in order to assess a minute but clinically meaningful rather than a significant growth.
Thank you for your consideration. We have integrated the sentence as suggested (Line 247-248)
Minor Comments/Clarifications with suggested alterations
Some points should be improved in terms of English language such as the following:
- Please note that “familiarity” probably does not express the intended meaning (Table 1 and Table 4). Please explain or rephrase.
Thank you for your suggestion, we have replaced “familiarity” with “hereditary” (Table 3 of the revised version). According to the indication of Reviewer 2, we have deleted the Table 1.
- Please note that the sentence:
“The mean mPTC growth rate in stable cases was 0.37 mm3/month while it was significantly greater in the mPTC which achieves a volume change ≥50% respect to the other” might be better rephrased as:
“The mean mPTC growth rate in stable cases was 0.37 mm3/month while it was significantly greater in the mPTC which achieves a volume change ≥50% with respect to the other”.
Thank you, we have modified as suggested (line 16)
Reviewer 2 Report
Active surveillance for papillary thyroid cancer is a field of interest to many readers. The content of this study is interesting, but it would be good to try more content supplementation. 1. The first 3 lines of the Result part seem unnecessary, so it is better to delete them. 2. In the result section, please add a table or clearly describe the number of people who have finally had thyroidectomy, the surgical method and the final pathology. I am also curious about how well the size of the cancer in the operated specimen coincided with the ultrasound measurements. 3. In the discussion, among molecular genetic tests for thyroid cancer, it would be good to mention tests that are attempted to predict or judge the progression of mPTC, and add their limitations and future prospects.Author Response
We thank the reviewer for their thoughtful criticism. In bold their comments and in italic our responses.
Active surveillance for papillary thyroid cancer is a field of interest to many readers. The content of this study is interesting, but it would be good to try more content supplementation.
- The first 3 lines of the Result part seem unnecessary, so it is better to delete them.
Thank you for your suggestion, we have deleted the first 2-3 lines and the first 2 tables, accordingly. Consequently, we have re-numbered the following tables (Lines 115, 120, 133, 136).
- In the result section, please add a table or clearly describe the number of people who have finally had thyroidectomy, the surgical method and the final pathology. I am also curious about how well the size of the cancer in the operated specimen coincided with the ultrasound measurements.
Thank you for the original point. We reported the data in a table (Table 1) and we cited it in the text (Lines 107-108)
- In the discussion, among molecular genetic tests for thyroid cancer, it would be good to mention tests that are attempted to predict or judge the progression of mPTC, and add their limitations and future prospects.
Thank you for underlining an important issue. On this regard we have addressed the issue in the discussion section of the revised manuscript, adding a sentence (lines 197-210) and two references.
Round 2
Reviewer 2 Report
I've verified that it's been properly revised for what was requested.I found the data in Table 1 very interesting.
Of the 109 AS group, only 5 patients progressed and underwent surgery, but 3 of them had lateral neck LN metastasis and 5 patients underwent total thyroidectomy.
Questions remain as to whether this subject could have been treated with more limited surgery instead of total thyroidectomy if it had been operated earlier.
In this study, the authors suggest that volume increase may be a better criterion for judging mPTC progression, but there is a limitation that the surgical results of subjects with volume increase cannot be referred to as that basis.
Although this study is insufficient setting to determine which of volume increase or diameter increase is the more appropriate evaluation criterion, considering the fact that it is difficult to design a study to perform surgery on all patients at the time the researcher wants,
it is difficult to realize in reality.
The authors suggested that there is room for recognition of the authors' argument in that a tighter standard than the existing evidence is needed because there is a disadvantage of missing the timing of limited surgery as the disease is considerably advanced when surgery is performed based on the criteria of is shown.
If such a description is added to the discussion part, it seems to be able to get more sympathy from readers.
Author Response
Dear Reviewer,
find below in bold you comment and in italic our responses.
I've verified that it's been properly revised for what was requested.
I found the data in Table 1 very interesting.
Of the 109 AS group, only 5 patients progressed and underwent surgery, but 3 of them had lateral neck LN metastasis and 5 patients underwent total thyroidectomy.
Questions remain as to whether this subject could have been treated with more limited surgery instead of total thyroidectomy if it had been operated earlier.
In this study, the authors suggest that volume increase may be a better criterion for judging mPTC progression, but there is a limitation that the surgical results of subjects with volume increase cannot be referred to as that basis.
Although this study is insufficient setting to determine which of volume increase or diameter increase is the more appropriate evaluation criterion, considering the fact that it is difficult to design a study to perform surgery on all patients at the time the researcher wants, it is difficult to realize in reality.
The authors suggested that there is room for recognition of the authors' argument in that a tighter standard than the existing evidence is needed because there is a disadvantage of missing the timing of limited surgery as the disease is considerably advanced when surgery is performed based on the criteria of is shown.
If such a description is added to the discussion part, it seems to be able to get more sympathy from readers.
Thank you for your point of view. We agree with you in saying that the delayed surgery could be more extensive than immediate surgery, but it is still safe in term of disease outcome and surgery adverse events, without additional disadvantage. Moreover, it is the only available strategy, that allows to operate the mPTC patients who needed a treatment, avoiding unnecessary surgery procedures for those patients who do not require it. It is relevant to highlighted that the percentage of patients who required to be treated in this period of about 2 years and half of is only 4.5% of all cases. The objective of future research will be to identify more and more early mPTC that tend to progress in order to define a timely and less extensive therapeutic approach. We add some comments in the discussion (lines 193-196 and 250-253)